# What Influences Proprioceptive Impairments in Patients with Rheumatic Diseases? Analysis of Different Factors

**DOI:** 10.3390/ijerph20043698

**Published:** 2023-02-19

**Authors:** Paweł Konarzewski, Urszula Konarzewska, Anna Kuryliszyn-Moskal, Robert Terlikowski, Jolanta Pauk, Kristina Daunoraviciene, Konrad Pauk, Agnieszka Dakowicz, Mariusz Wojciuk, Janusz Dzięcioł, Zofia Dziecioł-Anikiej

**Affiliations:** 1AXIS, Fabryczna 4/1d, 15-483 Białystok, Poland; 2Department of Rehabilitation, Faculty of Health Sciences, Medical University of Białystok, Skłodowskiej-Curie 7A Str., 15-096 Białystok, Poland; 3Institute of Biomedical Engineering, Faculty of Mechanical Engineering, Bialystok University of Technology, Wiejska 45C, 15-352 Białystok, Poland; 4Department of Biomechanical Engineering, Faculty of Mechanics, Vilnius Gediminas Technical University, Plytinės g. 25, LT-10105 Vilnius, Lithuania; 5Warsaw Medical University, Zwirki i Wigury 61, 02-091 Warsaw, Poland; 6Department of Human Anatomy, Faculty of Medicine, Medical University of Bialystok, Mickiewicza 2A Str., 15-230 Bialystok, Poland

**Keywords:** rheumatoid arthritis, osteoarthritis, postural disorders, proprioception

## Abstract

Rheumatic diseases lead to postural problems, which increase the risk of falls and lead to greater disability. The aim of the present work is to evaluate posture disorders in patients with osteoarthritis (OA) and rheumatoid arthritis (RA), as well as to evaluate the influence of other factors. A total of 71 subjects were enrolled in this study. Joint position sense (JPS) and the functional assessment of proprioception on a balance platform for both lower limbs were examined. The Average Trace Error (ATE), test time (t), and Average Platform Force Variation (AFV) were calculated. Additionally, an equilibrium test was carried out in the one-legged standing position (Single Leg Stance—SLS). The results were compared in several ways and revealed the following: (1) A JPS of 10° plantar flexion in RA obtained significantly worse results when repeating the movement than OA; the ATEs were significantly lower in RA; and RA needed more support during SLS assessment. (2) RA patients with higher DAS28 had statistically significantly higher values in JPS, with 5° plantar flexion and 10° dorsal flexion, SLS assessment, and stabilometric rates. A statistically significant correlation between DAS28 and RA was found in a JPS of 10° plantar flexion. The VAS ruler demonstrated a significant moderate correlation with t. (3) Patients who experienced at least one fall demonstrated higher JPS and t. Our study shows that proprioception is the most influenced by the nature of the disease and the level of disease activity. We can see that the stability and balance functions are also greatly influenced by the patient’s falling experience and the level of pain. These findings may be useful in designing an optimal proprioception-enhancing movement training plan.

## 1. Introduction

In the population of people over 65 years of age, falls are the leading cause of injuries [1,2,3,4,5,6,7,8,9,10,11]. This is due to elderly people having impaired proprioception, eyesight, vestibular system, muscle weakness, and decreased power and endurance [1,2,4,6,7,10,11]. The disturbances are also visible in the kinetic and kinematic characteristics of postural reactions, i.e., in their initiation, amplitude, and coordination of automatic responses. Deficits in the dynamic maintenance of postural control as a result of neuromuscular disorders observed in the elderly increase in the presence of diseases related to the locomotor system, such as osteoarthritis and rheumatoid arthritis [10,11,12]. In groups of patients with rheumatoid arthritis, over 30% of people report at least one fall within the last year [10,11], because RA affects the locomotor system [13,14,15,16,17]. The pathomechanism of articular changes in RA is related to the presence of rheumatoid granulation tissue, which penetrates the joint and causes the destruction of cartilage, bones, and the ligamentous apparatus. The characteristic symptoms are pain, swelling, limited mobility of the symmetrical joints of the hands and feet, and morning stiffness. Changes in the structure and balance of tensions within the musculoskeletal system lead to disturbances in dynamic synchronization and static stabilization mechanisms. Due to the large number of proprioceptors located within the damaged periarticular structures, the sensorimotor control is disturbed, which is why the static capacity of the feet is important for the control of posture, proper movement, and sensorimotor control [15,18]. In turn, about 50% of osteoarthritis patients report a fall at the same age. OA is a group of pathological changes resulting from the destabilization of the process of degradation and synthesis within the articular cartilage and the subcartilage layer of the bone [19,20,21,22,23]. The development of the disease leads to the narrowing of the joint space, subchondral sclerotization, and the formation of degenerative geodes. Structural damage leads to impaired sensorimotor function, which undoubtedly affects postural stability [3,7,9,12,19,22,24]. Although the symptoms of these two types of rheumatic diseases can be similar, it is very important to distinguish between them in order to determine the effect on proprioception and balance. Therefore, the aim of the study was to evaluate postural disorders in patients with rheumatic diseases: (1) by comparing the RA and OA patients, (2) by evaluating the influence of the DAS28 and VAS ruler, and (3) by relating fall experience and risk in patients.

## 2. Methods and Protocol

The study included 71 subjects (41 with RA and 30 with OA in lower limbs) treated at the Department of Rehabilitation, University Hospital in Białystok. RA was diagnosed according to ACR (American College of Rheumatology) and EULAR (European League Against Rheumatism) criteria from 2010 [25]. The evaluation of the severity of RA was conducted based on Steinbrocker’s classification [26], disease activity score—DAS 28 [25], and Gordon’s functional abilities score [27]. The evaluation of the progress of OA was conducted based on clinical and radiological evaluation. Medical history included: the duration of the disease, body mass and height, and the level of pain according to the Visual Analog Scale for Pain (VAS ruler) [28]. The VAS ruler is presented as a ruler with a movable cursor, the length being approximately 10 cm. “0” represents “No pain”, while “10” represents “Agonizing”. When the test began, subjects moved the cursor to the position on the ruler that best represented the degree of pain they were currently experiencing [29]. Patients excluded from the study suffered from co-existing conditions, which may also influence postural control: stage II clinical obesity, injuries within the lower limb in the last year, ophthalmic disorders which disturb the observation of markers on the screen during examination, dizziness of laryngological or neurological origin, paresis and paralysis, which disturb standing on two and one leg as well as moving the lower limb, proprioception disorders of neurological origin, cardiovascular diseases preventing moderate physical activity, diabetes, radiculopathy, and pregnancy. Patients gave written consent to participate in the study. The study was granted consent by the Bioethical Committee of Medical University of Białystok (nr: R-I-002/344/2013).

The examination of proprioception included the determination of joint position sense (JPS) and the functional assessment of proprioception on the balance platform for both lower limbs (Figure 1). During the joint position sense assessment, patients had to repeat three attempts without visual control. The angle position was previously presented by the examiner (5° and 10° of plantar flexion and dorsal flexion). The patient was in a standing position and one foot was set centrally on the mobile balance platform next to the other foot, which was set on the stable part of the platform. After the assessment of one lower limb, the position was changed to perform the assessment of the other lower limb (Figure 1).

The functional examination of proprioception was conducted using PRO KIN Line Software for the balance platform (TecnoBody model PK 254). The patients were in a standing position with the upper limbs resting on the hips; one foot was set centrally on the mobile balance platform next to the other foot, which was set on the stable part of the platform. After the assessment of one lower limb, the position was changed to perform the assessment of the other lower limb. During the examination, the patient made a circular movement in the range between 5° and 10° by controlling their foot on the platform. The app calculated Average Trace Error (ATE), test time in seconds, and Average Platform Force Variation (AFV). Additionally, an equilibrium test in the position of one-legged standing was carried out (Single Leg Stance-SLS), during which the number of supports per 30 s was calculated for both lower limbs.

## 3. Data Analysis and Statistical Evaluation

The comparison of the results of proprioception and SLS assessment were as follows: (1) Between RA and OA patients; JPS, ATE, and SLS comparison between study groups was performed by calculating the difference in means in RA and OA patients (difference of means = RA(mean_value)—OA(mean_value)). (2) Between groups of patients with moderate and high activity of RA; the group of patients with RA included patients with moderate (DAS28 < 5.1) and high (DAS28 > 5.1) activity of the disease. Patients with moderate and high activity of the disease according to DAS28 were compared regarding all measured variables. Means and SDs of JPS parameters were presented for individual groups and statistically evaluated with a confidence interval of 95%. Cohen’s effect size was calculated and presented along with the test duration. Cohen’s value *d* was calculated to evaluate the effect size: *d* = 0.01 (very small effect size), *d* = 0.20 (small effect size), *d* = 0.50 (medium effect size), *d* = 0.80 (large effect size), *d* = 1.20 (very large effect size), and *d* = 2.00 (huge effect size). The correlation analysis between the disease activity score DAS28, VAS ruler, and JPS, and SLS results was identified by using *r* Pearson and *rho* Spearman. A correlation coefficient of |r| < 0.3 was considered low, 0.3 ≤ |r| ≤ 0.7 was considered moderate, and |r| > 0.7 was considered high. (3) Depending on the number of falls within 1 year; comparative analysis was conducted on the group of patients who experienced at least one fall during in the last year in comparison to patients who did not experience falls regarding proprioception and SLS. The analysis was conducted separately in two study groups of patients. Statistical analyses were conducted using IBM SPSS Statistics 25. The *t*-test for independent samples, the Mann–Whitney U test, and one-way analysis of variance for dependent and independent samples were used for data analysis. Statistical significance was established as *p* < 0.05.

## 4. Results

### 4.1. Descriptive Analysis of Patient Groups

Groups of patients did not differ statistically significantly regarding age, BMI, duration of the disease, or level of pain in the VAS ruler (Table 1).

### 4.2. Comparison of JPS, ATE, and SLS in Study Groups

Analysis was conducted for JPS assessment in plantar and dorsal flexion movement of the ankle joint for RA and OA patients (Table 2).

Analysis indicates that in the case of 10° plantar flexion, patients with RA obtained a higher significant difference in repeating the movement than patients with OA. Statistically significant differences were demonstrated in the results of the functional examination of proprioception within ATE, which was lower in RA patients than in OA (Table 3).

It was proved that patients with RA used statistically significantly more supports during 30 s of SLS assessment in comparison to the group of patients with OA.

### 4.3. Comparison of Proprioception and SLS Results between Groups of Patients with Moderate and High Activity of RA

The results demonstrated that patients with a higher activity of the disease revealed statistically significantly higher values of error in JPS assessment with 5° plantar flexion (*p* < 0.05, and medium effects *d*  =  0.53) and 10° dorsal flexion in comparison to patients with a lower activity of the disease (*p* < 0.05 and medium effects *d*  =  0.73) (Table 4; Figure 2).

No statistically significant differences were observed within the results of functional examination of proprioception (ATE, AFV) between groups (*p* > 0.05) based on the RA activity level. A statistically significant difference was observed between patients in SLS assessment. Patients with higher activity of the disease revealed a higher number of supports in SLS assessment (4.61 ± 3.93 for patients with DAS28 < 5.1 vs. 7.04 ± 4.10 for patients with DAS28 > 5.1).

### 4.4. Correlation between DAS28, the VAS Ruler, and JPS and SLS Results

Results of the analysis show a statistically significant correlation of error of 10° plantar flexion mapping with the degree of RA activity measured with DAS28 (Table 5).

A statistically significant difference was found between the time of proprioception assessment completion and the VAS ruler (r = 0.41, *p* < 0.001), as well as changes in pressure on the stabilometric platform and the activity of the disease (r = 0.28, *p*< 0.05) in patients with RA. A statistically significant correlation was also shown between the number of supports in SLS assessment and RA activity (r = 0.34, *p* < 0.01).

### 4.5. Comparison of JPS and SLS Depending on the Number of Falls within 1 Year

It was proved in the RA group that the error within JPS assessment with 5° plantar flexion (*p* < 0.05, and medium effects *d*  =  0.46) and 10° dorsal flexion (*p* < 0.05 and medium effects *d*  =  0.50) was higher in patients who experienced at least one fall versus those who did not report falls. Similar findings were found in the OA group regarding the error within JPS assessment with 5° dorsal flexion (*p* < 0.05, and large effects *d*  > 0.80) and 10° dorsal flexion (*p* < 0.05 and medium effects *d*  =  0.50) (Table 6).

It was observed that in the group of patients with RA, the time of proprioception assessment completion was statistically significantly higher in subjects who reported at least one fall in the last year than those who did not report falls (*p* < 0.05, and medium effect size *d* = 0.61) (Table 7).

In patients with OA, a higher number of supports was observed in SLS assessment in patients with a history of falls in comparison to the group without falls (*p* < 0.05 and medium effect size *d* = 0.50). Differences were not found in the group of patients with RA.

## 5. Discussion

The objective of the study was to identify postural disorders in patients with RA and OA. For this purpose, assessments of JPS, functional proprioception, and SLS were conducted. Rheumatoid and degenerative processes have been shown to affect structures involved in proprioception [5,30,31,32]. Authors of other studies confirm the occurrence of proprioception disorders in patients with RA and OA in comparison to healthy controls [4,5]. We found that the rheumatoid and degenerative process affects structures related to proprioception, reducing the ability to sense joint position as a method of proprioception testing. The results indicated that the JPS of 10° plantar flexion in RA obtained significantly worse results in repeating the movement than OA (*p* = 0.002). Regarding the functional test of proprioception under visual control in the group of patients with RA, a lower Average Trace Error (ATE) and a shorter duration of the test were found compared to patients with OA. It was proved that such an assessment evaluates visual and proprioceptive components of the sensorimotor control system, which may influence the differences in relation to the results of JPS, which is performed without visual control [33,34,35]. This is consistent with [31,32]. The authors stated that an active inflammatory process can be a factor that enhances sensorimotor disorders, manifested by the greatest error in joint position sensing.

Cudejko et al. demonstrated a negative effect of active inflammation, determined on the basis of the erythrocyte sedimentation index on the JPS [36,37]. In our studies, a similar relationship was shown in RA patients when assessing the DAS28 index. In patients with a higher RA activity index, statistically significantly higher values of errors in mapping the range of motion were observed in the assessment of JPS within plantar flexion and dorsal flexion in comparison to those with a lower disease activity. The time of proprioception assessment correlated with pain based on the VAS ruler (r = 0.41, *p* < 0.001), and changes in pressure on the stabilometric platform correlated with the activity of the disease (r = 0.28, *p*< 0.05) in patients with RA. According to our research, the pain may decrease the sensitivity of the proprioceptive receptors, so the RA patients’ proprioception may be worse than that of healthy patients. Additionally, a statistically significant but moderate correlation was shown between the number of supports in SLS assessment and RA activity (r = 0.34, *p* < 0.01). The moderate correlation could be a result of different testing conditions compared to normal functions in daily activities. Patients with a higher activity of the disease revealed a higher number of supports in SLS assessment (4.61 ± 3.93 for patients with DAS28 < 5.1 vs. 7.04 ± 4.10 for patients with DAS28 > 5.1). According to Häkkinen et al. [38], patients with active RA have greater difficulties in performing activities related to postural control than patients with lower disease activity. The reports of other authors suggest that the severity of the rheumatoid process may cause disturbances in maintaining postural control due to exacerbation of disease symptoms that have a negative impact on proprioception and other sensorimotor functions [39,40].

The etiology of falls is multifactorial and may result from a complex interaction of internal, environmental, and behavioral components. Falls may lead to loss of mobility, injury, and death. The increased risk of falls in patients with RA and OA is associated with the presence of pain, joint deformities, limited physical activity and muscle strength, and disturbances in gait and postural control. Proprioception disorders as a component of the postural control system lead to an increased risk of falls [41,42,43,44,45]. In our study, 42.0% of patients with RA and 41.4% of patients with OA had a fall history within the last year, and this rate is in agreement with other studies. For example, in [46], 35% of RA patients had a fall history within the last year. A similar rate of falling was observed by Armstrong [47]; it was 33%. The present study confirms the results of studies by other authors, showing that in both groups, patients with at least one fall had more position mapping errors in the assessment of joint position sense compared to those who did not report falls in medical history. Moreover, it was found that in the group of patients with RA, the time to perform a functional proprioception test was statistically significantly higher in those who reported at least one fall during the year (*p* < 0.05). Moreover, the error within JPS assessment with 5° plantar flexion (*p* < 0.05, and medium effects *d*  =  0.46) and 10° dorsal flexion (*p* < 0.05 and medium effects *d*  =  0.50) was higher in patients who did report falls. Similar results were found in the OA group regarding the error within JPS assessment with 5° dorsal flexion (*p* < 0.05, and large effects *d*  > 0.80) and 10° dorsal flexion (*p* < 0.05 and medium effects *d*  =  0.50). Similar results were observed in OA patients. Moreover, a higher number of supports in SLS assessment were observed in OA patients with a history of falls in comparison to the group without falls (*p* < 0.05 and medium effect size *d* = 0.50). Zhou et al. also found a relationship between proprioception and postural control disorders and the risk of falls in elderly subjects [48,49,50,51]. Moreover, the authors of [46] showed decreased strength and proprioception in rheumatic patients, which impair postural stability. The limitation of our study is the lack of analysis based on gender. Further research should focus on clarification of the cause–effect relationship.

We believe that research in this direction needs to be continued, but its most important limitations should be considered. Our research can be extended not only to gender-based analysis, but also to age-based analysis. In order to confirm the findings of this study, more methods of quantitative and qualitative assessment of proprioception should be included, thus including more causes, such as neuropathies, which are common and characteristic of RA and its factors. Additionally, observing the obtained results leads to the idea of applying stricter selection criteria.

## 6. Conclusions

The analysis identified several factors with a positive contribution to the correlation between the time of proprioception assessment and pain, disability activity, and number of supports in RA patients. These findings might be useful in the design of optimal proprioception-enhancing movement training.

## Figures and Tables

**Figure 1 ijerph-20-03698-f001:**
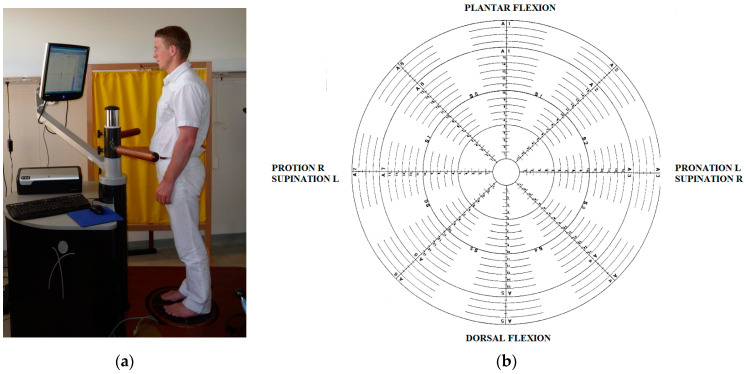
Measurement protocol: (**a**) examination on stabilometric platform; (**b**) TecnoBody PK 254 stabilometric platform structure.

**Figure 2 ijerph-20-03698-f002:**
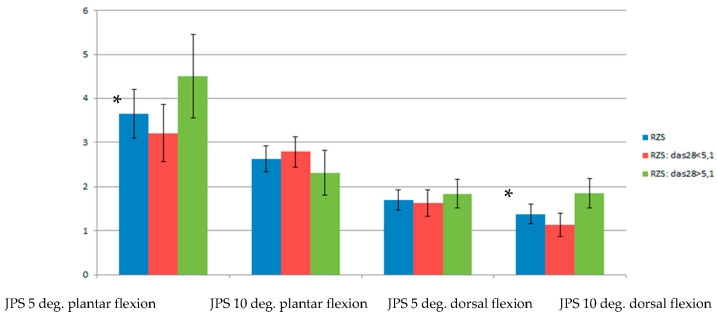
Comparison of chosen results of JPS assessment between RA groups: blue color column represents all patients, red column represents patients with moderate activity of disease, green represents patients with high activity of the disease, * *p* < 0.05.

**Table 1 ijerph-20-03698-t001:** Descriptive parameters of study groups.

	Age (Years)		Duration of the Disease (Years)		BMI (kg/m^2^)		VAS Ruler (mm)	
	*M*	*SD*	*M*	*SD*	*M*	*SD*	*M*	*SD*
**RA**	61.8	9.7	13.3	8.7	26.8	3.7	52.5	21.6
**OA**	63.0	9.9	10.9	5.9	27.8	3.6	47.6	15.7

Values represented as mean (*M*) with standard deviation (*SD*).

**Table 2 ijerph-20-03698-t002:** Comparison of JPS in plantar and dorsal flexion motion of ankle joint between groups.

	RA vs. OA	
	Difference of Means	*p*
JPS 5 deg. plantar flexion—mean absolute error	−0.17	0.90
JPS 5 deg. plantar flexion—mean relative error	−0.03	0.91
JPS 5 deg. dorsal flexion—results range in samples	−0.37	0.42
JPS 10 deg. plantar flexion—mean absolute error	0.49	0.02
JPS 10 deg. plantar flexion—mean relative error	0.05	0.05
JPS 10 deg. plantar flexion—results range in samples	0.16	0.66
JPS 5 deg. dorsal flexion—mean absolute error	−0.17	1.00
JPS 5 deg. dorsal flexion—mean relative error	−0.03	1.00
JPS 5 deg. dorsal flexion—results range in samples	0.00	1.00
JPS 10 deg. dorsal flexion—mean absolute error	0.01	0.99
JPS 10 deg. dorsal flexion—mean relative error	0.00	0.99
JPS 10 deg. dorsal flexion—results range in samples	0.10	0.62

**Table 3 ijerph-20-03698-t003:** Comparison of proprioception functional examination results in a circular movement between groups.

	RA vs. OA	
	Difference of Means	*p*
ATE (%)	−6.48	0.02
ATE (time)	2.43	0.77
AFV (kg)	−0.22	0.23

**Table 4 ijerph-20-03698-t004:** Comparison of the results of JPS assessment between groups for both lower limbs based on the activity of the disease.

	RA: DAS28 < 5.1 (*n* = 54)		RA: DAS28 > 5.1 (*n* = 28)				95% *CI*		
	*M*	*SD*	*M*	*SD*	*t, sec.*	*p*	*LL*	*UL*	*d*Cohen
JPS 5 deg. plantar flexion—mean absolute error	3.21	2.42	4.51	2.53	−2.27	0.02	−2.44	−0.16	0.53
JPS 5 deg. plantar flexion—mean relative error	0.64	0.48	0.90	0.50	−2.26	0.02	−0.48	−0.03	0.53
JPS 5 deg. plantar flexion—range of results in attempts	1.84	1.29	1.95	1.80	−0.33	0.74	−0.80	0.57	0.08
JPS 10 deg. plantar flexion—mean absolute error	2.79	1.27	2.31	1.35	1.57	0.11	−0.13	1.08	0.37
JPS 10 deg. plantar flexion—mean relative error	0.28	0.13	0.23	0.13	1.57	0.11	−0.01	0.11	0.37
JPS 10 deg. plantar flexion—range of results in attempts	1.75	1.19	1.46	1.28	1.02	0.31	−0.28	0.85	0.24
JPS 5 deg. dorsal flexion—mean absolute error	1.63	1.12	1.84	0.86	−0.89	0.37	−0.70	0.27	0.21
JPS 5 deg. dorsal flexion—mean relative error	0.33	0.22	0.37	0.17	−0.80	0.42	−0.13	0.06	0.19
JPS 5 deg. dorsal flexion—range of results in attempts	0.99	0.75	1.13	0.90	−0.74	0.46	−0.51	0.24	0.17
JPS 10 deg. dorsal flexion—mean absolute error	1.14	0.99	1.85	0.90	−3.16	0.00	−1.15	−0.26	0.73
JPS 10 deg. dorsal flexion—mean relative error	0.11	0.10	0.18	0.09	−3.16	0.00	−0.12	−0.03	0.73
JPS 10 deg. dorsal flexion—range of results in attempts	0.73	0.59	0.78	0.59	−0.30	0.76	−0.31	0.23	0.07

Values represented as mean (*M*) with standard deviation (*SD*), *LL* stands for lower level and *UL* stands for upper level of confidence interval.

**Table 5 ijerph-20-03698-t005:** Correlation coefficient *r* between DAS28, VAS ruler, and JPS in RA and OA groups.

	RA—DAS28	RA—VAS	OA—VAS Ruler
JPS 5 deg. plantar flexion—mean absolute error	0.13	0.11	0.13
JPS 5 deg. plantar flexion—mean relative error	0.13	0.11	0.13
JPS 5 deg. plantar flexion—range of results in attempts	0.08	0.09	−0.08
JPS 10 deg. plantar flexion—mean absolute error	−0.16	0.09	−0.04
JPS 10 deg. plantar flexion—mean relative error	−0.16	0.09	−0.04
JPS 10 deg. plantar flexion—range of results in attempts	−0.05	−0.03	0.02
JPS 5 deg. dorsal flexion—mean absolute error	0.03	0.01	0.12
JPS 5 deg. dorsal flexion—mean relative error	0.02	0.00	0.11
JPS 5 deg. dorsal flexion—range of results in attempts	−0.05	0.08	0.08
JPS 10 deg. dorsal flexion—mean absolute error	0.24 *	0.04	−0.15
JPS 10 deg. dorsal flexion—mean relative error	0.24 *	0.04	−0.15
JPS 10 deg. dorsal flexion—range of results in attempts	0.03	0.08	−0.21

* *p* < 0.05.

**Table 6 ijerph-20-03698-t006:** Comparison between results of JPS between patients with falls and without falls in the group with RA and with OA.

**RA**									
	No falls within last year (*n* = 48)		Falls within last year (*n* = 34)				95% *CI*		
	*M*	*SD*	*M*	*SD*	*t, sec.*	*p*	*LL*	*UL*	*d*Cohen
JPS 5 deg. plantar flexion—mean absolute error	3.18	2.32	4.31	2.67	−2.04	0.045	−2.23	−0.03	0.46
JPS 5 deg. plantar flexion—mean relative error	0.64	0.46	0.86	0.53	−2.05	0.044	−0.45	−0.01	0.46
JPS 5 deg. plantar flexion—range of results in attempts	1.82	1.43	1.96	1.55	−0.40	0.690	−0.79	0.53	0.09
JPS 10 deg. plantar flexion—mean absolute error	2.56	1.36	2.72	1.25	−0.54	0.594	−0.75	0.43	0.12
JPS 10 deg. plantar flexion—mean relative error	0.26	0.14	0.27	0.12	−0.54	0.594	−0.07	0.04	0.12
JPS 10 deg. plantar flexion—range of results in attempts	1.73	1.25	1.53	1.18	0.72	0.475	−0.35	0.74	0.16
JPS 5 deg. dorsal flexion—mean absolute error	1.64	1.02	1.79	1.08	−0.68	0.499	−0.62	0.31	0.15
JPS 5 deg. dorsal flexion—mean relative error	0.33	0.20	0.36	0.22	−0.66	0.509	−0.12	0.06	0.15
JPS 5 deg. dorsal flexion—range of results in attempts	0.99	0.63	1.12	1.00	−0.67	0.506	−0.52	0.26	0.16
JPS 10 deg. dorsal flexion—mean absolute error	1.18	0.96	1.67	1.02	−2.25	0.027	−0.94	−0.06	0.50
JPS 10 deg. dorsal flexion—mean relative error	0.12	0.10	0.17	0.10	−2.25	0.027	−0.09	−0.01	0.50
JPS 10 deg. dorsal flexion—range of results in attempts	0.53	0.34	1.05	0.72	−3.89	<0.001	−0.78	−0.25	0.97
**OA**									
	No falls within last year (*n* = 34)		Falls within last year (*n* = 24)				95% *CI*		
	*M*	*SD*	*M*	*SD*	*t, sec.*	*p*	*LL*	*UL*	*d*Cohen
JPS 5 deg. plantar flexion—mean absolute error	3.55	2.51	4.36	2.23	−1.26	0.212	−2.09	0.47	0.34
JPS 5 deg. plantar flexion—mean relative error	0.71	0.50	0.87	0.44	−1.26	0.214	−0.42	0.10	0.34
JPS 5 deg. plantar flexion—range of results in attempts	2.10	1.50	2.52	2.34	−0.77	0.444	−1.52	0.68	0.22
JPS 10 deg. plantar flexion—mean absolute error	2.21	1.36	2.16	0.98	0.13	0.894	−0.61	0.70	0.04
JPS 10 deg. plantar flexion—mean relative error	0.22	0.14	0.22	0.10	0.13	0.894	−0.06	0.07	0.04
JPS 10 deg. plantar flexion—range of results in attempts	1.23	0.87	1.85	1.18	−2.30	0.025	−1.16	−0.08	0.61
JPS 5 deg. dorsal flexion—mean absolute error	1.52	0.80	2.43	1.38	−2.91	0.006	−1.55	−0.28	0.85
JPS 5 deg. dorsal flexion—mean relative error	0.31	0.16	0.48	0.28	−2.81	0.008	−0.30	−0.05	0.82
JPS 5 deg. dorsal flexion—range of results in attempts	1.02	0.80	1.08	0.90	−0.29	0.777	−0.52	0.39	0.08
JPS 10 deg. dorsal flexion—mean absolute error	1.18	0.83	1.73	1.21	−2.05	0.045	−1.08	−0.01	0.55
JPS 10 deg. dorsal flexion—mean relative error	0.12	0.08	0.17	0.12	−2.05	0.045	−0.11	0.00	0.55
JPS 10 deg. dorsal flexion—range of results in attempts	0.69	0.75	0.59	0.50	0.60	0.550	−0.25	0.46	0.16

Values represented as mean (*M*) with standard deviation (*SD*); *LL* stands for lower level and *UL* stands for upper level of confidence interval.

**Table 7 ijerph-20-03698-t007:** Comparison of the results of functional proprioception assessment between groups with and without falls with RA and OA.

**RA Patients**									
	No falls within last year (*n* = 48)		Falls within last year (*n* = 34)				95% *CI*		
	*M*	*SD*	*M*	*SD*	*t, sec.*	*p*	*LL*	*UL*	*d*Cohen
ATE (%)	28.35	10.32	29.59	7.48	−0.60	0.554	−5.36	2.89	0.13
ATE (time)	76.81	20.39	89.59	21.43	−2.74	0.008	−22.07	−3.49	0.61
AFV (kg)	1.02	0.76	0.88	0.62	0.86	0.391	−0.19	0.47	0.20
**OA patients**									
	No falls within last year (*n* = 34)		Falls within last year (*n* = 24)				95% *CI*		
	*M*	*SD*	*M*	*SD*	*t, sec.*	*p*	*LL*	*UL*	*d*Cohen
ATE (%)	32.24	12.26	40.33	20.57	−1.72	0.094	−17.64	1.44	0.50
ATE (time)	80.56	22.16	78.63	17.09	0.36	0.721	−8.87	12.74	0.10
AFV (kg)	1.03	0.77	1.38	0.85	−1.65	0.105	−0.78	0.08	0.44

Values represented as mean (*M*) with standard deviation (*SD*); *LL* stands for lower level and *UL* stands for upper level of confidence interval.

## Data Availability

The data presented in this study are available on request from the corresponding author (zofia.dzieciol-anikiej@umb.edu.pl).

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
