# Peer review of "What Influences Proprioceptive Impairments in Patients with Rheumatic Diseases? Analysis of Different Factors"

_ijerph, 2023, doi:10.3390/ijerph20043698_

Round 1

Reviewer 1 Report

Review of the article entitled “What influences proprioceptive impairments in patients with rheumatic 2 diseases? Analysis of diferent factors”

The authors took up a very interesting research topic. The purpose of this study was to the aim of the study was to evaluate postural disorders in patients with rheumatic diseases.

I have no comments on the Introduction. A short introduction provides a clear background to the work. Properly captured the importance of the problem. In the introduction, quite new literature items were cited.

The bibliography contains a sufficient number of articles -  50 literature items, which proves the in-depth study of the issues raised in the existing literature. The cited literature has been well selected.

The methodology has been correctly described in the article the studied group is representative.

The study groups were correctly selected, the groups did not differ in terms of age, BMI, etc.

The results were well presented, correctly conducted statistical analyzes - selected statistical methods are correct and sufficient. The presentation of the results obtained is clear. Most of the results are presented in tables.

The discussion was conducted correctly.

My minor comments:

1) Fig. 2 should contain a legend clearly showing what results are presented by the successive colors of the bars.

2) Is the only limitation of the work is the lack of gender-based analyses? Please provide more limitations and possible directions for further research at the end of the discussion.

3) Of course, the form of the article should be corrected, in accordance with the guidelines of the journal.

4) The method of citing articles in the text should be improved - in accordance with the order in which the article / book appears in the text - in accordance with the guidelines of the journal.

I recommend the article be published in the International Journal of Environmental Research and Public Health journal after making minor corrections listed above.

Author Response

First of all, we thank the reviewer for his highly positive evaluation of our scientific publication and for pointing out the missing information in the manuscript. We assure you that we have done everything to improve the publication.

Point 1: Fig. 2 should contain a legend clearly showing what results are presented by the successive colors of the bars.

Response 1:  Thank you very much. We corrected Figs. 2 according to your comment.

Point 2: Is the only limitation of the work is the lack of gender-based analyses? Please provide more limitations and possible directions for further research at the end of the discussion.

Response 2:  We thank the reviewer for the comment and we provided limitation in the discussion section end. We’ve marked it in red.

Point 3: Of course, the form of the article should be corrected, in accordance with the guidelines of the journal.

Response 3:  We thank the reviewer for the comment. We have made formatting improvements based on the journal requirements.

Point 4: The method of citing articles in the text should be improved - in accordance with the order in which the article / book appears in the text - in accordance with the guidelines of the journal.

Response 4: Thanks for the note. We have provided the list of references according to the requirements of the journal.

Reviewer 2 Report

Over all a good effort to detect the problems in propioception in RA and OA 

these are the following objections? 

Why two different diseases were selected ?

Authors didnt ruled out neuropathy which is prevalent in long standing cases of RA ? the problems in Propioception might be due to neuropathy?

Author Response

We thank the reviewer for his interest in this topic and kind reflection. We made improvements to the manuscript based on comments. We hope and trust that all reviewers' comments and concerns have been fully and satisfactorily addressed. Detailed responses to comments follow.

Point 1: These are the following objections. Why two different diseases were selected?

Response 1:  Thank you for your note. However, we have indicated that we chose these two different diseases for research because they are both linked by posture disorders. We explained our choice in the Introduction. “Although the symptoms of these two types of rheumatic diseases can be similar, it is very important to distinguish between them in order to determine the effect on pro-prioception and balance. Therefore, the aim of the study was to evaluate postural dis-orders in patients with rheumatic diseases: (1) by comparing RA and OA patients, (2) by evaluating the influence of DAS28 and VAS ruler, and (3) by relating fall experience and risk in patients”.

Point 2: Authors didn’t ruled out neuropathy which is prevalent in long standing cases of RA? The problems in Propioception might be due to neuropathy?

Response 2:  We agree with the observation and plan to evaluate neuropathy in future studies. We did not evaluate the influence of neuropathy, as this was not foreseen in the research methodology, we strictly followed the plan. Furthermore, this is one of the limitations of the study and we have indicated it along with other observations at the end of the discussion (highlighted in red).

Round 2

Reviewer 2 Report

Dear Authors 

your Introduction section is way to lengthy it will make readers bored if write the each and every aspect o disease you should write those things of disease which are only related to your topic, its not necessary to introduce the disease whole for disease.

in this study sentences are too lengthy to understand and lot of medical jargon is there which needs to be rectified 

Methodology section is also very much confusing and the special tests which are applied are not explained rather more focused is on commentary about other aspects

Results;  in this section data is presented in large number which may be shortlisted and only important and related tables may be incorporated 

Over all this article needs lot editing to cut the sentences , so a readers sholdn't get bored while reading this article 

Author Response

We thank the reviewer for his interest in this topic and kind reflection. We made improvements to the manuscript based on comments. We hope and trust that all reviewers' comments and concerns have been fully and satisfactorily addressed. Detailed responses to comments follow.

Point 1: Your Introduction section is way to lengthy it will make readers bored if write the each and every aspect o disease you should write those things of disease which are only related to your topic, its not necessary to introduce the disease whole for disease.

Response 1:  Thanks for the note. The introduction has been shortened (Please see attachment).

Point 2: In this study sentences are too lengthy to understand and lot of medical jargon is there which needs to be rectified

Response 2:  Thanks for the note. The earlier version of the article was, as suggested by the reviewer, subjected to linguistic proofreading by qualified persons from MDPI (certificate attached).

Point 3: Methodology section is also very much confusing and the special tests which are applied are not explained rather more focused is on commentary about other aspects

Response 3:  Thanks for the note. The tests described in the methodology used in the study are international tests and have been described in accordance with the world classification. The entire work concept, including the detailed methodology, was developed by a team led by a professor specializing in rheumatology and medical rehabilitation.

Point 4: Results;  in this section data is presented in large number which may be shortlisted and only important and related tables may be incorporated

Response 4: Referring to the comments related to the large number of results, they are caused by the previously adopted methodology. only the most significant results were included in the paper.

Over all this article needs lot editing to cut the sentences , so a readers sholdn't get bored while reading this article

Thank you for your comments. Bearing in mind the changes we have made, we hope that they will make the article more readable.
